

# Relationship between swimming speed, intra-cycle variation of horizontal speed, and Froude efficiency during consecutive stroke cycles in adolescent swimmers

Mafalda P. Pinto[1,2], Daniel A. Marinho[1,2], Henrique P. Neiva[1,2] and Jorge E. Morais[2,3]

[1] Department of Sport Sciences, University of Beira Interior, Covilhã, Portugal
[2] Research Centre in Sports, Health and Human Development (CIDESD), Covilhã, Portugal
[3] Department of Sport Sciences, Instituto Politécnico de Bragança, Bragança, Portugal

## ABSTRACT

The purpose of this study was to investigate the relationship between swimming speed, intra-cycle variation of horizontal speed of displacement ($dv$), and Froude efficiency ($\eta_F$) in front-crawl during three consecutive stroke cycles. The sample consisted of 15 boys aged 16.07 ± 0.77 years and 15 girls aged 15.05 ± 1.07 years. Swimming speed, $dv$ and $\eta_F$ were measured during a 25 m front-crawl trial. Three consecutive stroke cycles were measured. Swimming speed showed a non-significant stroke-by-stroke effect (F = 2.55, $p$ = 0.087, $\eta^2$ = 0.08), but a significant sex effect (F = 90.46, $p$ < 0.001, $\eta^2$ = 0.76). The $dv$ and $\eta_F$ had the same trend as the swimming speed for the stroke-by-stroke effect, but a non-significant sex effect ($p$ > 0.05). The Spearman correlation matrix between swimming speed and $dv$, and swimming speed and $\eta_F$ showed non-significant correlations for the three stroke cycles in both sexes. However, the tendency of the former was not always inverse, and the latter was not always direct. Coaches and swimmers need to be aware that lower $dv$s are not always associated with faster swimming speeds and *vice-versa*, and that $\eta_F$ is a predictor of swimming speed, not $dv$.

## INTRODUCTION

Competitive swimming is a cyclical sport in which swimmers attempt to cover a given distance in the shortest possible time. Of the four swimming strokes (front-crawl, backstroke, breaststroke, and butterfly), the front-crawl has received a great deal of attention from researchers and coaches because it is the fastest, the one most often used in training, and the one that most swimmers master (*Maglischo, 2003*; *Morais et al., 2022a*). This stroke is characterized by being a ventral (body position), alternating (upper and lower limb action) and continuous (force production) stroke (*Maglischo, 2003*). Thus, to achieve better performances, swimmers must increase (or maintain) propulsion while minimizing the hydrodynamic resistance opposite to their displacement (*Toussaint & Beek, 1992*).

Corresponding author
Mafalda P. Pinto,
mafpamplonapinto@gmail.com

However, during the front-crawl stroke (as well as in other swimming strokes), the interaction of propulsion and drag forces creates body accelerations and decelerations, resulting in a variation in the swimmer's center of mass (*Ribeiro et al., 2013*; *Gonjo et al., 2021*). This variation is usually referred to as the "intra-cycle variation of horizontal displacement velocity" (*dv*) (*Barbosa et al., 2005*). For convenience, researchers often calculate the *dv* as the coefficient of variation (CV = (one standard deviation)/mean * 100, in %) (*Barbosa et al., 2005*; *Figueiredo et al., 2016*; *Silva et al., 2019*). The *dv* is considered an indicator of swimming efficiency (*Figueiredo et al., 2012*; *Ribeiro et al., 2013*). Lower values tend to indicate a higher gross propulsive efficiency (for the same drag condition) (*Peterson Silveira et al., 2019*). Thus, research in swimming has shown a great interest in the inter- and intra-variability of the *dv* (*Barbosa et al., 2013b*; *Figueiredo et al., 2013*; *Fernandes et al., 2022b*). The former refers to between-subject comparisons, while the latter refers to within-subject comparisons. Because it is a variable that is easily measured by researchers and efficiently interpreted by coaches and swimmers, researchers are interested in providing information about the *dv*. The greatest variation in swim speed detected indicates that swimmers tend to perform a greater acceleration/deceleration pattern. On the other hand, a smoother swim speed–time curve indicates that the swimmers tend to decelerate less and thus maintain a more constant swim speed (*Ganzevles et al., 2019*).

However, the literature suggests that the results provided by the *dv* may be misleading. Some studies have reported that lower *dv* is associated with better performance (*Barbosa et al., 2013b*; *Matsuda et al., 2014*). Nonetheless, it has also been shown that the *dv* does not have a linear relationship with performance in a lap-by-lap analysis (*Psycharakis et al., 2010*), and there were no significant associations between the *dv* and swimming performance (*Ruiz-Navarro, Morouço & Arellano, 2020*). These conflicting data may be explained by differences in the drag profile or the efficiency parameters of the swimmers, and therefore may not be directly related to the speed variation itself (*Fernandes et al., 2022a*). The front-crawl stroke (as well as other strokes) shows a high variability (*Bideault, Herault & Seifert, 2013*). That is, swimmers tend to show a meaningful difference in their stroke mechanics between stroke cycles during the same trial (*Morais et al., 2020*). This is a common phenomenon observed in sports. A study by *Preatoni et al. (2013)* showed that when a movement is performed repetitively, there is some variability in the body's motion, even in cyclical movements. In the case of swimming, this variability may be even greater due to the characteristics of the aquatic environment (*e.g.*, density and viscosity). Therefore, different swimming strokes may show different patterns. The most commonly used method to measure *dv* is to take the average of a series of stroke cycles during a trial (*e.g.*, *Barbosa et al., 2014*; *Figueiredo et al., 2016*). However, as far as is known, only *Fernandes et al. (2022b)* attempted to gain insight into the relationship between swimming speed and *dv* in a stroke-by-stroke analysis. Based on two consecutive paired stroke cycles, the authors argued that the *dv* observed in one stroke cycle does not influence the variation of the subsequent stroke cycle, nor is it related to the mean speed (*Fernandes et al., 2022b*). Thus, there is a lack of evidence in the literature for a stroke-by-stroke analysis that can provide a deeper insight into the relationship between swimming speed and *dv*.

**Table 1 Characteristics (mean ± one standard deviation—SD) of the swimmers by sex.**

| | Mean ± SD | |
| --- | --- | --- |
| | **Males (N = 15)** | **Females (N = 15)** |
| Age (years) | 16.07 ± 0.77 | 15.05 ± 1.07 |
| Body mass (kg) | 67.53 ± 5.69 | 56.55 ± 6.14 |
| Height (m) | 1.76 ± 0.05 | 1.63 ± 0.07 |
| Arm span (m) | 1.82 ± 0.09 | 1.67 ± 0.07 |
| FINA points (points) | 581.40 ± 62.37 | 596.27 ± 76.14 |

Given that $dv$ results from the interaction of various swim limitations and that it differs depending on the level of swimming performance (*Barbosa et al., 2019*; *Fernandes et al., 2022b*), examining more consecutive stroke cycles would provide a deeper understanding of the effect of the $dv$ on swimming speed. In addition, other efficiency variables have been shown to be good indicators of swimmer efficiency, such as the Froude efficiency ($\eta_F$) (*Zamparo et al., 2005*). Indeed, it has been reported that young swimmers with greater swimming efficiency ($\eta_F$) are more likely to have better swimming performances (*Barbosa et al., 2019*). Therefore, this study aimed to: (i) understand the change in swimming speed, $dv$, and $\eta_F$ over three consecutive stroke cycles by sex, and; (ii) understand the relationship between the swimming speed, $dv$, and $\eta_F$ in front-crawl during three consecutive stroke cycles in a stroke-by-stroke analysis. It was hypothesized that swimming speed, $dv$ and $\eta_F$ would show a significant change over time, with swimming speed and $\eta_F$ decreasing and the $dv$ increasing. Furthermore, a significant relationship between swimming speed and both efficiency variables would be verified. More specifically, lower $dv$s (inverse relationship) and higher $\eta_F$ (direct relationship) would lead to faster swimming speeds.

## MATERIALS AND METHODS

### Participants

The sample consisted of 30 adolescent swimmers (15 males and 15 females). The characteristics of the swimmers are presented in Table 1. Their performance level (FINA points) was calculated based on the 100 m freestyle short-course event. They were recruited from a national team that regularly participated in regional, national and international competitions. The sample included age-group national record holders, age-group national champions, and other swimmers who enrolled in national talent identification programs (Tier 3 athletes) (*McKay et al., 2021*). They trained six to nine times a week. At the time of data collection, they were in peak form at the end of the second macro-cycle. Inclusion criteria were that the swimmers should be front-crawl experts and have no limitations (*e.g.*, no injuries in the past 6 months) that would prevent them from performing at their best. Participants were provided with verbal and written information about the study and then provided written informed consent to participate in the study. All procedures were in accordance with the Declaration of Helsinki for research involving
human subjects, and the research was approved by the Ethics Committee of the Instituto Politécnico de Bragança (N.° 72/2022).

## Experimental design

Prior to data collection, swimmers performed a standardized warm-up for sprint swimming (*Neiva et al., 2017*). Then, after an auditory signal, the swimmers were asked to perform three all-out 25-m front-crawl trials with a push-off start, with a 30-min rest between each trial. The best trial (fastest swimming speed) was used for further analysis. Three consecutive stroke cycles after the 10 m mark were analyzed. This was done to avoid any advantage of the wall push-off. Swimmers were instructed to perform non-breathing stroke cycles during the data collection section to avoid changes in stroke coordination or technique that could negatively affect swimming speed (*McCabe, Sanders & Psycharakis, 2015*).

## Measurement of swimming speed and speed fluctuation (*dv*), and Froude efficiency ($\eta_F$)

To measure the swimming speed, the string of a mechanical device (SpeedRT; ApLab, Rome, Italy) was attached to the waist of the swimmers (*Morais et al., 2022b*). The speedometer calculated the swimmer's displacement and speed at a rate of 100 Hz. The speed–time series were then imported into a signal processing software (AcqKnowledge v. 3.9.0; Biopac Systems, Santa Barbara, Goleta, CA, USA). After residual analysis (*Winter, 2009*), the signal was handled with a 4th order Butterworth low-pass filter (cut-off frequency: 5 Hz). A video camera (GoPro Hero Black 7; GoPro, San Mateo, CA, USA) was placed in a fixed position in the mid-section of the swimming pool. It was synchronized with the mechanical device to record the swimmers in the sagittal plane and to identify the entry and exit of the hands into the water. Swimming speed (in m/s) was obtained from the software during three consecutive stroke cycles (based on hand entry and exit). The *dv* (in %) of each stroke cycle was then calculated as the coefficient of variation (CV): CV = one standard deviation/mean * 100 (*Barbosa et al., 2005*).

The Froude efficiency ($\eta_F$) was calculated as:

$$\eta_F = \left( \frac{v \cdot 0.9}{2\pi \cdot SF \cdot l} \right) \cdot \frac{2}{\pi} \cdot 100$$

where $\eta_F$ is the Froude efficiency (in %), v is the swimming speed (in m/s), SF is the stroke frequency (in Hz), and l is the shoulder-to-hand average distance (in m) (*Zamparo et al., 2005*). Stroke frequency was measured by calculating the number of cycles per unit of time from the time required to complete a full cycle (f = 1/t), and then converted to Hz. The variable l was measured on land using digital photogrammetry (*Morais et al., 2020*).

## Statistical analysis

An *a priori* power analysis was performed using G*Power (*Faul et al., 2009*). For a repeated-measure analysis, 10 participants were required to detect a moderate effect size (d = 0.50) with 80% power ($\alpha$ = 0.05) with two groups and three measurements for a

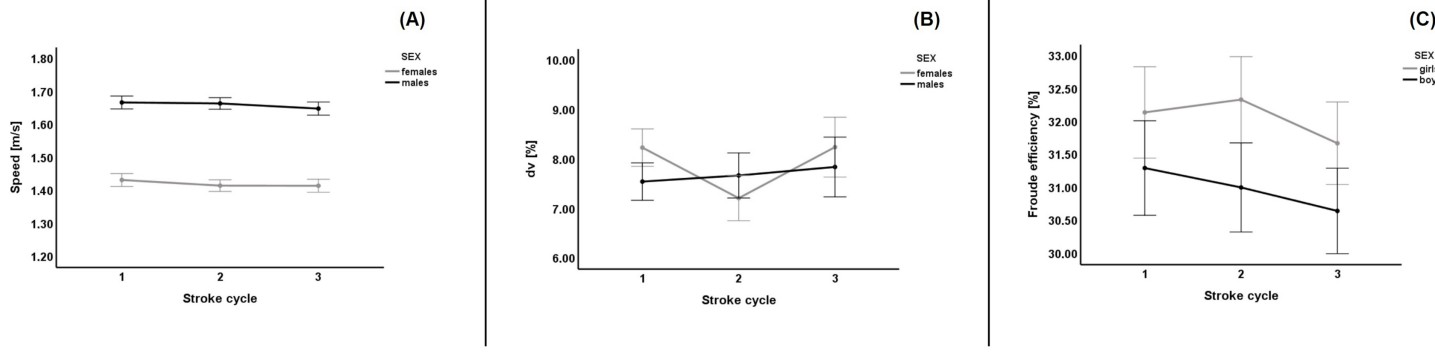

**Figure 1 Swimming speed, intra-cycle variation of the horizontal speed and Froude efficiency analysis by time and sex.** (A) Swimming speed; (B) intra-cycle variation of the horizontal speed of displacement (dv); (C) Froude efficiency ($\eta_F$). 1, first stroke cycle; 2, second stroke cycle; 3, stroke cycle. Bars represent one standard error.           

repeated measures, within- and between-subjects ANOVA statistical test.

The Shapiro-Wilk test was used to assess the sample distribution. Swimming speed, *dv* and $\eta_F$ showed a normal distribution. The mean plus one standard deviation was calculated as descriptive statistics.

Two-way repeated measures ANOVA was used to verify changes over time (*i.e.*, differences between the three consecutive stroke cycles) ($\alpha = 0.05$). Thus, two factors were considered: stroke-by-stroke and sex, and their interaction. The Bonferroni correction was used to detect significant pairwise differences when necessary. The effect size index (eta square—$\eta^2$) was calculated and interpreted as: (i) no effect if $0 < \eta^2 \leq 0.04$; (ii) minimal if $0.04 < \eta^2 \leq 0.25$; (iii) moderate if $0.25 < \eta^2 \leq 0.64$; and (iv) strong if $\eta^2 > 0.64$ (*Ferguson, 2009*). The Spearman correlation coefficient ($p < 0.05$) was used to verify the correlation between swimming speed and *dv*, and $\eta_F$. This was done to assess the strength and direction of the correlations between pairs of variables.

The multilevel hierarchical linear model (HLM) procedure was used to test whether swimming speed could be predicted by the *dv* and $\eta_F$, or both. This is a statistical regression model designed to account for the hierarchical or nested structure of the data. In the first level, variables that allowed repeated measures (*dv* and $\eta_F$) were tested as predictors. Time, considered as the inter-cycle change between the first and third stroke cycles, was also tested as a predictor. In the second level, variables that did not change (sex) were also tested as predictors. Only significant predictors were included in the final model. Maximum likelihood estimation was performed using HLM7 software (*Raudenbush et al., 2011*).

## RESULTS

Figure 1 shows the descriptive data of swimming speed, *dv*, and $\eta_F$ by sex during the three consecutive stroke cycles. Male swimmers were faster than female swimmers in all stroke cycles (Fig. 1). Both sexes showed a similar trend of decreasing swimming speed over time. The *dv* tended to increase in males and decrease and increase in females. As for the $\eta_F$, it tended to decrease over time. Table 2 shows the data related to the analysis of variance. A non-significant stroke-by-stroke effect with a minimal effect size was observed for

**Table 2 ANOVA repeated measures of the swimming speed and *dv* and $\eta_F$.**

|  | F-ratio | *p*-value | $\eta^2$ |
|---|---|---|---|
| Speed (m/s) | | | |
| Stroke-by-stroke effect | 2.55 | 0.087 | 0.08 |
| Sex effect | 90.46 | <0.001 | 0.76 |
| Stroke-by-stroke X Sex interaction | 0.57 | 0.571 | 0.02 |
| *dv* (%) | | | |
| Stroke-by-stroke effect | 1.17 | 0.318 | 0.04 |
| Sex effect | 0.17 | 0.681 | 0.01 |
| Stroke-by-stroke X Sex interaction | 1.06 | 0.352 | 0.04 |
| $\eta_F$ (%) | | | |
| Stroke-by-stroke effect | 2.05 | 0.139 | 0.07 |
| Sex effect | 1.48 | 0.235 | 0.05 |
| Stroke-by-stroke X Sex interaction | 0.33 | 0.723 | 0.02 |

**Note:**
*dv*, speed fluctuation; $\eta_F$, Froude efficiency; $\eta^2$, eta square (effect size index).

**Table 3 Spearman correlation matrix between swimming speed, *dv* and $\eta_F$ in each stroke cycle by sex.**

|  | Males | | | Females | | |
|---|---|---|---|---|---|---|
|  | Speed[1st] | Speed[2nd] | Speed[3rd] | Speed[1st] | Speed[2nd] | Speed[3rd] |
| *dv*[1st] | $r_s = -0.050$<br>$p = 0.860$ | | | $r_s = 0.164$<br>$p = 0.558$ | | |
| *dv*[2nd] | | $r_s = -0.229$<br>$p = 0.413$ | | | $r_s = -0.061$<br>$p = 0.830$ | |
| *dv*[3rd] | | | $r_s = 0.325$<br>$p = 0.237$ | | | $r_s = 0.214$<br>$p = 0.443$ |
| $\eta_F^{1st}$ | $r_s = 0.116$<br>$p = 0.692$ | | | $r_s = -0.125$<br>$p = 0.657$ | | |
| $\eta_F^{2nd}$ | | $r_s = -0.178$<br>$p = 0.543$ | | | $r_s = -0.243$<br>$p = 0.383$ | |
| $\eta_F^{3rd}$ | | | $r_s = 0.114$<br>$p = 0.697$ | | | $r_s = -0.032$<br>$p = 0.909$ |

**Note:**
*dv*, intra-cycle variation of the horizontal speed of displacement. $\eta_F$, Froude efficiency; 1st, first stroke cycle measured; 2nd, second stroke cycle measured; 3rd, third stroke cycle measured.

swimming speed. Conversely, a significant sex effect with a strong effect size was observed. A non-significant stroke-by-stroke X sex interaction was observed. Non-significant effects (stroke-by-stroke, sex, and the respective interaction) were found for the *dv* and $\eta_F$. This indicates that the *dv* did not change significantly over time and that there were no significant differences between the sexes.

Table 3 shows the Spearman correlation matrix between swimming speed, *dv*, and $\eta_F$ in each stroke cycle. Because a significant sex effect was observed for swimming speed, the correlations between swimming speed, *dv*, and $\eta_F$ are also presented by sex. In males, the

**Table 4 Fixed effects of the final models computed with standard errors (SE) and 95% confidence intervals (95% CI).**

| Fixed effect | Estimate (SE) | 95% CI | $p$ |
|---|---|---|---|
| Sex | 0.242 (0.027) | [0.189–0.295] | <0.001 |
| $\eta_F$ | 0.010 (0.003) | [0.004–0.016] | 0.002 |

Note:
$\eta_F$, Froude efficiency.

Spearman correlation coefficient showed an inverse relationship (*i.e.*, higher $dv$ led to slower swimming speeds) between swimming speed and $dv$ in the first and second stroke cycles. However, these correlations were not significant in all stroke cycles. As for $\eta_F$, it was direct in the first and third stroke cycles, but inverse in the second, with non-significant correlations. In girls, as in boys, the correlations were not significant over all stroke cycles measured. Furthermore, only the second stroke cycle was inversely related. Curiously, the correlations between swimming speed and $\eta_F$ were inverse in all stroke cycles, but not significant. Thus, the correlation between $dv$ and swimming speed was not always inverse, and the correlation between $\eta_F$ and swimming speed was not always direct, but always non-significant.

Table 4 presents the significant predictors of swimming speed. Time (*i.e.*, the intercycle change between the first and third stroke cycles) was not retained as a significant predictor. This means that although the swimming speed decreased over time, it was not significant. The final model retained sex as a significant predictor indicating differences between the sexes. Instead of $dv$, $\eta_F$ was retained as a significant predictor of swimming speed. For each unit increase in $\eta_F$ (in %), the swimming speed increased by 0.010 m/s.

## DISCUSSION

The purpose of this study was to understand the relationship between the swimming speed, intra-cycle variation of the horizontal speed of displacement ($dv$), and Froude efficiency ($\eta_F$) in front-crawl during three consecutive stroke cycles. The main results of this study showed that all variables had a non-significant stroke-by-stroke effect. On the other hand, swimming speed showed a significant sex effect, but $dv$ and $\eta_F$ did not. Regarding the correlations, non-significant correlations were found between swimming speed and $dv$, and $\eta_F$ in both sexes. The relationship was not always inverse for the $dv$ (lower $dv$ led to faster swimming speeds), and direct for the $\eta_F$ (greater $\eta_F$ led to faster swimming speeds). In addition, HLM indicated $\eta_F$ as a predictor of swimming speed, but not the $dv$.

In the present study, a stroke-by-stoke analysis of swimming speed and $dv$, and $\eta_F$ was performed instead of an average of the whole trial. Studies have shown that sprint swimmers tend to decrease their swimming speed over time during an event/trial (*Morais et al., 2020*, *2022b*). The present results, which included both male and female adolescent swimmers, are consistent with these findings (decrease over time), but not significantly so. As for the swimming speed, the $dv$ did not show a significant variance. For males, an inverse trend was observed. That is, $dv$ increased as the swimming speed decreased.

For females, instead of an increase over time (based on the rationale that lower $dv$ leads to faster swimming speeds and *vice-versa*), a sinusoidal profile was observed. It was expected that the $dv$ would show a significant change over time, increasing as swimming speed decreased. The $dv$ is a popular indicator of swimming efficiency with a direct positive effect on energy cost (*Barbosa et al., 2006*). That is, in all four swimming strokes, swimming with a lower $dv$ can induce a decrease in the energy cost of the swimmer to cover a given distance (*Barbosa et al., 2011*). In cross-sectional studies, analysis of maximal trials in front-crawl indicated that faster swimming speeds were associated with lower $dv$ in young swimmers (*Barbosa et al., 2013a*; *Silva et al., 2019*). When analyzing national-level swimmers between the ages of 11 and 13 over the course of a season, swimming speed decreased from the first to the second assessment moment and increased from the second to the third, presenting a sinusoidal profile (*Morais et al., 2013*). On the other hand, $dv$ showed the opposite trend (*i.e.*, when swimming speed decreased between the evaluation moments, $dv$ increased and *vice-versa*). This trend was also observed when the swimmers were divided into clusters based on their performance level (*Morais et al., 2015*). The three performance clusters showed a similar profile (*i.e.*, when swimming speed decreased between assessment moments, $dv$ increased and *vice-versa*) (*Morais et al., 2015*). Even after the instructional program, swimmers improved their performance by increasing their swimming speed while decreasing their $dv$ (*Costa et al., 2017*; *Silva et al., 2022*). Thus, from a general perspective, the literature suggests that there is a strong and inverse relationship between swimming speed and $dv$, which was only partially confirmed by the results of the current study.

In the present study, $\eta_F$ showed the same results as the $dv$, *i.e.*, non-significant stroke-by-stroke and sex effects, but with a tendency to decrease over time. This indicates that swimming speed may not always have an inverse relationship with the $dv$, especially in females. Conversely, swimming speed seems to have a direct relationship with $\eta_F$. Furthermore, the present results show that the correlations between swimming speed and $dv$, and swimming speed and $\eta_F$ were not significant in either sex. Regarding the relationship between swimming speed and $dv$, the trend was not always inverse in both sexes (lower $dv$ leading to faster swimming speed and *vice-versa*). In fact, there are studies reporting opposite results on the relationship between swimming speed and $dv$ over time, but not in consecutive strokes as in the present study. For example, Barbosa and collaborators verified that young swimmers did not significantly change their $dv$ during a three-evaluation moment program (*Barbosa et al., 2015*). These findings were also reported in another study (*Morais et al., 2017*). The authors observed that during three competitive seasons (10 assessment moments), there was an overall trend for young swimmers to increase their swimming speed (except during the detraining period). However, the $dv$ remained almost unchanged during the same period (*Morais et al., 2017*). The $\eta_F$ was also monitored in the same study (*Morais et al., 2017*). Although increases and decreases were observed within each competitive season, the $\eta_F$ increased from the first to the third competitive season. Others have analyzed the relationship between swimming speed and $dv$ in national-level adult swimmers (*Psycharakis et al., 2010*). However, instead of a 25 m maximal trial, they analyzed this relationship during a 200 m maximal trial in a

lap-by-lap analysis. The authors found that the $dv$ did not have a linear relationship with performance and did not change significantly during the 200 m trial (*Psycharakis et al., 2010*). However, all of these studies mentioned above share a common factor, *i.e.*, the swimming speed and $dv$ were analyzed as an average of the trial (or by lap in the case of the 200 m trials). Therefore, the average may not accurately represent each stroke cycle performed. Sport performance in general (*Preatoni et al., 2013*), and swimming in particular, has been reported to exhibit both inter- and intra-variability (*Bideault, Herault & Seifert, 2013*). Specifically, stroke-by-stroke variability may occur in swimming, where athletes have to perform movements under unstable conditions (*Simbana-Escobar, Hellard & Seifert, 2018*). Therefore, to better understand the relationship between swimming speed and its determinants, a stroke-by-stroke analysis may provide deeper insights into this topic.

Overall, this study presents conflicting results regarding the relationship between swimming speed and $dv$ in a stroke-by-stroke analysis. In males, swimming speed and $dv$ showed an inverse trend (swimming speed decreased and $dv$ increased). In females, swimming speed decreased but the $dv$ showed a sinusoidal profile. In addition, non-significant correlations were found between swimming speed and $dv$ in all three stroke cycles measured in both sexes. The correlations were not always inverse for both sexes. That is, in some stroke cycles, $dv$ had a direct correlation with swimming speed (*i.e.*, greater $dv$ led to faster swimming speeds). Additionally, and using multilevel modeling (HLM), $dv$ was not retained as a significant predictor of swimming speed. Recently, it has been reported that the $dv$ may not be the cause of a behavior, but rather a consequence (*Fernandes et al., 2022b*, *2023*). Swimmers can increase their swimming speed in two ways: (i) by generating greater propulsive forces in a more powerful manner (resulting in lower efficiency), or; (ii) by demonstrating greater swimming efficiency (*Peterson Silveira et al., 2019*). Therefore, the $dv$ can be seen as a consequence of the motor behavior performed. The first approach will result in a higher $dv$ (more pronounced speed fluctuations), and the second will result in a lower $dv$ (smoother speed fluctuations). These findings, as well as those of other studies, suggest that the $dv$ may not always have an inverse and meaningful relationship with swimming speed. Conversely, the $\eta_F$ based on multilevel modeling was retained as a significant predictor of swimming speed with a significant and direct effect (greater $\eta_F$ leads to faster swimming speeds). Nevertheless, the $dv$ continues to provide an overall perspective on swimmer efficiency, especially in youth swimming (*Barbosa et al., 2019*), and in teaching contexts (*Bartolomeu et al., 2023*). However, a stroke-by-stroke analysis is recommended to better understand the relationship between swimming speed and the $dv$, especially in elite swimmers who may have multiple and different ways to improve their swimming performance. Coaches must know and understand the stroke mechanics and motor behavior of their swimmers by monitoring them with a large variety of tests related to this topic. This will allow them to hypothetically recognize that a higher $dv$ may not be a negative factor, but rather an intrinsic consequence of the swimmer's profile. As a result, they may be able to identify other variables (such as the $\eta_F$) that may be more important to the improvement of their swimmers. This emphasizes the individual

factor in swimming performance, *i.e.*, each swimmer is a unique individual and may have different ways of improving his/her performance.

The main limitations of this study are that (i) these findings apply to adolescent swimmers and front-crawl sprint trials/events; (ii) only three stroke cycles were analyzed, and; (iii) a coordinative index, such as the index of coordination (*Schnitzler, Seifert & Button, 2021*), was not used to gain deeper insight into the relationship between swimming speed and $dv$. Therefore, future studies should consider understanding these relationships in different age groups, different swimming strokes, and different racing speeds. Also, because sprint swimming is characterized by a significant decrease in speed over time in all four swimming strokes (*Morais et al., 2022a*), measuring more stroke cycles during the event/trial, may provide more insightful information about these relationships. Understanding the relationship between swimming speed, $dv$, and the coordination index may also provide important information on this topic. This is because the coordination index measures the time gap between the propulsive phases of each upper limb, which may have a significant effect on the $dv$ and consequently on the swimming speed. Therefore, future studies should be conducted to obtain more information on the relationship between swimming speed and $dv$, especially based on a stroke-by-stroke analysis.

## CONCLUSIONS

The results of this study showed that adolescent sprint swimmers presented a decrease in swimming speed over time, but without a significant stroke-by-stroke effect. Conversely, a sex effect was observed, with boys swimming faster than girls in all measured stroke cycles. For the $dv$ and $\eta_F$, non-significant stroke-by-stroke and sex effects were found. The correlations between swimming speed and $dv$ and swimming speed and $\eta_F$ were not significant in all three stroke cycles. Moreover, there were mixed results regarding the trend of the relationship. Instead of the $dv$, the $\eta_F$ was retained as a significant predictor of swimming speed. Thus, it can be concluded that the decrease in swimming speed over time was not significantly related to the increase in $dv$ in either sex.

### Funding

This work is supported by national funds (FCT—Portuguese Foundation for Science and Technology) under the project UIDB/DTP/04045/2020. The funders had no role in study design, data collection and analysis, decision to publish, or preparation of the manuscript.

### Grant Disclosures

The following grant information was disclosed by the authors:
FCT—Portuguese Foundation for Science and Technology: UIDB/DTP/04045/2020.

### Competing Interests

The authors declare that they have no competing interests.

## Author Contributions

- Mafalda P. Pinto performed the experiments, prepared figures and/or tables, and approved the final draft.
- Daniel A. Marinho conceived and designed the experiments, performed the experiments, prepared figures and/or tables, authored or reviewed drafts of the article, and approved the final draft.
- Henrique P. Neiva conceived and designed the experiments, performed the experiments, analyzed the data, authored or reviewed drafts of the article, and approved the final draft.
- Jorge E. Morais conceived and designed the experiments, performed the experiments, analyzed the data, prepared figures and/or tables, authored or reviewed drafts of the article, and approved the final draft.

## Human Ethics

The following information was supplied relating to ethical approvals (*i.e.*, approving body and any reference numbers):

Ethics Committee of the Instituto Politécnico de Bragança approved the research (N.° 72/2022).

## Data Availability

The raw measurements are available in a Supplemental File.

## Supplemental Information

Supplemental information for this article can be found online at http://dx.doi.org/10.7717/peerj.16019#supplemental-information.

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
