# Peer review of "Relationship between swimming speed, intra-cycle variation of horizontal speed, and Froude efficiency during consecutive stroke cycles in adolescent swimmers"

_PeerJ, doi:10.7717/peerj.16019_

## Round 0.1 · original submission · Minor Revisions

The paper is interesting and well-written. However, authors should follow the reviewers' suggestions.

Reviewer 1 ·

Basic reporting

The paper is well-written and concise.

Experimental design

Overall, the methods section is very well presented.
The aims are within the scope of the journal and the research question is relevant and meaningful.
The methods section is described with detail.

However, I would like to know better this question:
- The age of the age group swimmers, as used juniores and juvenile.
- Please add some comments on the breathing pattern during the tests and how this can affect performance.

Validity of the findings

The main findings are well reported and in connection with the methods.
However, I would like to know more about the use of 3 strokes.
- Please also add some more practical concerns and implications for coaches and swimmers.
- Please add explicitly the main limitations of the paper.

Additional comments

--

Reviewer 2 ·

Basic reporting

The authors tried to understand velocity fluctuation (intracycle velocity variation, IVV) in front crawl according to different competitive levels/tiers and swimmers’ sex. From a general point of view, the paper is very interesting and within the scope of the journal. I read the manuscript with great eagerness. I found the idea really interesting, with a big field to explore. Although the paper is very good, there are some concerns to be addressed to improve the quality of the paper
.
The introduction is well written and concise. However, I would recommend highlighting the main aim of the paper and the hypothesis of the study, probably with some rewriting of this section.

Experimental design

Regarding the methods section, I found odd the swimmers’ age in each group (due to the mean and standard deviation reported) and the statistical section is not clear enough to me. Could you please clarify this concern?

Moreover, I miss some information about the protocol (e.g., warm-up, rest between trials,…) and the justification of some of the procedures conducted (e.g., why three strokes?). Please address also this content

Validity of the findings

Finally, regarding the result and discussion sections, please add some more practical implications for coaches and swimmers as a coach-friendly concept.

Additional comments

Please also add some specific information on the limitations of the paper and the consequences for this analysis and further projects under this field

·

Basic reporting

Dear Authors,
The topic is relevant and although very specific and technical, the style is clear and it has a good readability.

Some specific comments:

Abstract
First line: the aim .. was to investigate the..
third line: in front-crawl during three consecutive..
"The Spearman correlation matrix between swimming speed and dv, and swimming speed and ·F revealed non-signifcant correlations for the three.."

Introduction
Line 58: research in swimming...

Material and methods
Line 102: Swimmers' characteristics...
Line 111: please clarify this sentence
Line 121: Do you mean that prior to the recorded trial? Please be clearer.
Line 157: … two-way was used...
Lines 193-194: please be clearer “one again…”

Results
Please try to be more coincise and synthetic, as you have several tables included in the paper.

Discussion
Line 216: … during three consecutive…
Line 223: The dv is a popular indicator…
Lines 223-246: in these paragraphs the studies reported should be more linked and referred to your results, or, vice versa. If not, it sounds more like a second Introduction section.
Lines 247-284: as above reported, there are several studies but presented as a new introduction part. Please, start from your results and link to previous contrasting or similar studies.

Experimental design

As reported in the study limitation section, do you think then that measuring more
strokes during the same event/trial would be more effective in understanding this relation?
Can you explain better this point?

Validity of the findings

No comment

Additional comments

I suggest to carefully review the language style trying to avoid as much as possible colloquial expressions and preferring an impersonal form.

---

## Round 0.2 · accepted · Accept

Authors have addressed all the reviewers' comments and suggestions as I personally checked. Therefore, the present version can be accepted for publication.